# Retrospective Analysis of the Outcome of Hospitalized COVID-19 Patients with Coexisting Metabolic Syndrome and HIV Using Multinomial Logistic Regression

**DOI:** 10.3390/ijerph20105799

**Published:** 2023-05-12

**Authors:** Peter M. Mphekgwana, Musa E. Sono-Setati, Tania V. Mokgophi, Yehenew G. Kifle, Sphiwe Madiba, Perpetua Modjadji

**Affiliations:** 1Research Administration and Development, University of Limpopo, Polokwane 0727, South Africa; 2Department of Public Health Medicine, University of Limpopo, Private Bag X1106, Sovenga, Polokwane 0727, South Africa; musasono@yahoo.com; 3Department of Statistics and Operations Research, University of Limpopo, Polokwane 0727, South Africa; taniamokgophi@gmail.com; 4Department of Mathematics and Statistics, University of Maryland Baltimore County, Baltimore, MD 21250, USA; yehenew@umbc.edu; 5Faculty of Health Sciences, University of Limpopo, Polokwane 0727, South Africa; sphiwe.madiba@ul.ac.za; 6Non-Communicable Diseases Research Unit, South African Medical Research Council, Cape Town 7505, South Africa; perpetua.modjadji@mrc.ac.za

**Keywords:** metabolic syndrome, non-communicable diseases, HIV, COVID-19, hospitalization, South Africa

## Abstract

Globally, the coexistence of metabolic syndrome (MetS) and HIV has become an important public health problem, putting coronavirus disease 19 (COVID-19) hospitalized patients at risk for severe manifestations and higher mortality. A retrospective cross-sectional analysis was conducted to identify factors and determine their relationships with hospitalization outcomes for COVID-19 patients using secondary data from the Department of Health in Limpopo Province, South Africa. The study included 15,151 patient clinical records of laboratory-confirmed COVID-19 cases. Data on MetS was extracted in the form of a cluster of metabolic factors. These included abdominal obesity, high blood pressure, and impaired fasting glucose captured on an information sheet. Spatial distribution of mortality among patients was observed; overall (21–33%), hypertension (32–43%), diabetes (34–47%), and HIV (31–45%). A multinomial logistic regression model was applied to identify factors and determine their relationships with hospitalization outcomes for COVID-19 patients. Mortality among COVID-19 patients was associated with being older (≥50+ years), male, and HIV positive. Having hypertension and diabetes reduced the duration from admission to death. Being transferred from a primary health facility (PHC) to a referral hospital among COVID-19 patients was associated with ventilation and less chance of being transferred to another health facility when having HIV plus MetS. Patients with MetS had a higher mortality rate within seven days of hospitalization, followed by those with obesity as an individual component. MetS and its components such as hypertension, diabetes, and obesity should be considered a composite predictor of COVID-19 fatal outcomes, mostly, increased risk of mortality. The study increases our understanding of the common contributing variables to severe manifestations and a greater mortality risk among COVID-19 hospitalized patients by investigating the influence of MetS, its components, and HIV coexistence. Prevention remains the mainstay for both communicable and non-communicable diseases. The findings underscore the need for improvement of critical care resources across South Africa.

## 1. Introduction

After the emergence of a novel coronavirus (Severe Acute Respiratory Syndrome-CoronaVirus-2 (SARS-CoV-2) in China, a pandemic spread worldwide causing 527,857,044 cases and nearly 6,301,101 casualties [1,2]. Africa, like other regions affected by the pandemic, recorded an increase in the number of hospitalized adult patients with suspected or confirmed COVID-19. An observation study of adults admitted to critical care in 10 countries in Africa recorded a death rate of 48%, which was higher than the global average of 31.5% [3]. South Africa has endured four waves of COVID-19 with various strains. To date, South Africa has approximately 4,063,237 confirmed cases, with an estimated 102,595 COVID-19-related deaths, and 3,912,506 recovered [4]. Progress with vaccination rollout has recorded 38 million vaccine doses that have been administered [5]. Based on epidemiological data, the risk of severe presentations and worse clinical outcomes of COVID-19 has been associated with pre-existing conditions such as cardiovascular disease, diabetes, hypertension, and overweight/obesity [6,7]. In Italy, the most prevalent comorbidities among patients with confirmed COVID-19 were hypertension and diabetes [8]. However, in other countries, diabetes did not pose an increased risk of being infected with SARS-CoV-2. Thus, very few cohort studies in Wuhan (China) [9] and the USA [10] have reported the relationship between hypertension, obesity, and diabetes with COVID-19 outcomes. In South Africa, early data associated COVID-19 morbidity and mortality with conditions of metabolic syndrome (MetS) as key factors in addition to respiratory failure [11]. This was consistent with initial studies of COVID-19 in Europe that identified respiratory failure as the leading cause of morbidity and mortality among hospitalized patients [11].

Globally, patients with MetS have a greater risk of contracting the COVID-19 virus and a poorer prognosis than patients without MetS component diseases [6,7]. MetS is defined by a coexistence of conditions that occur simultaneously increasing the risk for future cardiovascular diseases. The conditions include elevated blood pressure, elevated blood glucose, excess waist circumference, subcutaneous abdominal fat, and abnormal cholesterol or triglyceride levels [11]. MetS predisposes to morbidity and mortality in the general population [12,13]. Patients with MetS who co-present with these inherently overlapping risk factors may be at even higher risk of developing severe viral disease than patients with one of these conditions alone. The development of MetS blunts the host immune response to viral infection by influencing the immune system in various ways [14].

South Africa continues to experience the burden of non-communicable diseases (NCDs) and risk factors [14,15,16]. Health facility data show that uncontrolled and undiagnosed hypertension is especially prevalent among adults attending primary health facilities (PHC) [17,18] across semi-urban and rural settings [19,20]. The country also reports a high prevalence of diabetes, with a high proportion (38%) of people with undiagnosed diabetes [21]. A recent population-based study reported a prevalence of diabetes of 28% [22]. Excess waist circumference and subcutaneous abdominal fat are metabolic comorbidities that contribute to the burden of NCDs in South Africa, and the prevalence of overweight/obesity ranges between 56–68% [23,24].

In South Africa, 5% to 62% of adults suffer from MetS, depending on the subgroup that is studied [25,26,27,28,29]. Although research is minimal on its association with COVID-19, provincial data on COVID-19 fatalities suggest that people with diabetes are more likely to die from COVID-19 than any other high-risk group [30]. In a USA study, adult patients of African origin with MetS were 3.4 times more likely to die from COVID-19 than those who did not have the condition. Patients with MetS were also nearly five times more likely to be admitted to an ICU, need a ventilator, or develop acute respiratory distress syndrome [31].

When the co-presence of COVID-19 and NCDs was investigated, hypertension was associated with an increased risk of admission to the hospital and high mortality in patients infected with COVID-19 [32]. Increased affinity and susceptibility for SARS-CoV-2 among people with hypertension is due to altered expression of the angiotensin-converting enzyme-2 (ACE2) receptor [33,34,35]. However, a pathophysiological mechanism explaining the association between MetS and COVID-19 has not been established [31]. A possible mechanism suggests that the pre-existing endothelial dysfunction observed at an early stage of atherosclerosis in patients with MetS may play a crucial role in the development of severe COVID-19 [36]. Among overweight/obese individuals, susceptibility to pathology by SARS-CoV-2 could be due to reduced nitric oxide and other anti-inflammatory mechanisms.

MetS coexisting with HIV is an important public health problem in the African region, including South Africa [16,37,38]. During the COVID-19 pandemic, MetS was associated with severe manifestations of COVID-19 symptoms and higher mortality from COVID-19. Therefore, the coexistence of MetS and HIV increases the risk of severe COVID disease among people living with HIV [39]. This is particularly true of South Africa, given the high prevalence of HIV across population groups. Around 8 million of South Africa’s 60 million population live with HIV [40], affecting various population groups [41,42,43]. Immunocompromised people who are at greater risk of prolonged infection are vulnerable to COVID-19 and potentially more likely to host mutations of SARS-CoV-2 [44,45,46]. The risk of COVID-19 may be lower in HIV-infected individuals taking anti-HIV treatment than in the general population.

The treatment options and the management of patients hospitalized with COVID-19 are still being studied [47]. Some antiviral treatment medications, such as ivermectin, which has recently received FDA approval, chloroquine, remdesivir, ribavirin, and lopinavir/ritonavir, among others, have been used to treat individuals who have SARS-CoV-2 infections. However, an effective and specific anti-SARS-CoV-2 drug is still to be identified [48] On the other hand, retrospective data from studies conducted in Italy show that early hospitalization and early use of remdesivir may reduce COVID-19 progression to more severe respiratory disease and provide clinical benefit in patients with COVID-19 [48,49,50].

Considering the paucity of data on MetS and HIV coexisting among hospitalized COVID-19 patients and the view that this coexistence has become an important public health problem worldwide, including in South Africa, putting COVID-19 hospitalized patients with severe manifestations at higher mortality risk, this study aimed to identify factors and determine their relationships with hospitalization outcomes for COVID-19 patients using a secondary data from the Department of Health in the Limpopo Province, South Africa.

## 2. Materials and Methods

### 2.1. Study Design and Setting

A retrospective cross-sectional study was conducted between March 2020 to September 2022 in the Limpopo Province, South Africa. Limpopo is one of the nine provinces in South Africa, with its headquarters situated at Polokwane, located a few kilometres from the University of Limpopo in the Capricorn district. The province has five districts, namely (from largest to smallest population); Vhembe, Capricorn, Sekhukhune, Mopani, and Waterberg. The overall estimated total population of Limpopo province is 6 million, with over 1 million households and a Human Development Index of 0.710, which is regarded as the third highest in South Africa. Over three-quarters of the province is made up of rural localities and has the majority of black people, mostly earning their livelihood by subsistence farming or working as migrant labourers, and livestock raising is widespread [51]. 

This study used secondary data kept by the Limpopo Department of Health (LDoH). Ethical issues and permission to access the data are described in the published protocol [52], which was approved by Turfloop Research Ethics Committee (TREC) at UL and was granted (TREC/293/2021: IR). The study included 15,151 patient clinical records of laboratory-confirmed COVID-19 cases. Of the total records with complete information, Capricorn district contributed 5773, Waterberg (2496), Vhembe (2446), Mopani (2807), and Sekhukhune (1628). Only authorized researchers and provincial managers were given access to the entire database of hospitalized COVID-19 patients, which was maintained by the LDoH. In accordance with the Protection of Personal Information (POPI) Act of South Africa and the Declaration of Helsinki, all ethical norms of research pertaining to patient data were upheld.

The umbrella study further collected MetS in a form of a cluster of metabolic factors including abdominal obesity, high blood pressure, and impaired fasting glucose [18]. Obesity was regarded as a body mass index (BMI) greater than (≥30 kg/m^2^). A blood pressure of 120/80 mmHg was considered normal and greater or equal to 130/85 mmHg was classified as hypertension. Glucose level equal to or above 11.1 mmol/L or being on treatment for control of blood glucose is diagnosed as diabetes. HIV seropositive status was self-reported by the patient [18]. 

### 2.2. Statistical Analysis

A multinomial logistic regression model was applied to identify factors and determine their relationships with hospitalization outcomes for COVID-19 patients. In the model, hospitalization outcomes were the target variable and risk factors were explanatory variables (gender, age category, smoking status, hypertension, diabetes, obesity, HIV status, oxygenation, and ventilation). SPSS version 26.0 (IBM SPSS Statistics) (IBM Corp, Armonk, NY, USA) was used to perform descriptive analysis (frequency, percentages, and cross-tabulation), and the Chi-squared test to compare sets of nominal data that had larger frequency counts while Fisher’s exact test was used when frequency cells were small (less than five or ten). A Mantel–Haenszel extension test for trend was used to assess whether there was a linear trend in the hospitalization outcomes. The multinomial logistic regression analysis was carried out using the STATA software (Stata 9.0, StataCorp, College Station, TX, USA). All the maps were produced using ArcGIS version 10.8. 2. The Kaplan–Meier curves were utilized to compare the time to mortality between independent groups (hypertension (38% hypertensive vs. 62% non-hypertensive), diabetes (29% diabetic vs. 71% non-diabetic), obesity (11% obese vs. 89% non-obese), and MetS (4% MetS vs. 96% non-MetS)). The Log-Rank test was used as an inferential test to assess if there is a significant difference between the independent groups in their time–to mortality. 

## 3. Results

A total of 15151 COVID-19 cases were reported from March 2020 to September 2022. Mortality rates among hospitalized COVID-19 patients ranges were observed overall (20.47–32.62%), by hypertension (31.59–43.05%), diabetes (33.88–46.45%), and HIV (30.72–44.87%). In general, the spatial analysis indicated an unbalanced geographical distribution of mortality rate among hospitalized COVID-19 patients in Limpopo province, with a high mortality rate in the South part of the province. The Sekhukhune district had the highest incidence among the five districts, followed by the Vhembe district. In terms of mortality rate by hypertension and HIV among hospitalized COVID-19 patients, the Capricorn district had the highest incidence rate, followed by Vhembe and Sekhukhune districts. For mortality rate by diabetes among hospitalized COVID-19 patients, Sekhukhune district had the highest incidence, followed by Capricorn and Vhembe districts, shown in Figure 1. 

Thirty-eight hospitalized COVID-19 patients were hypertensive and 29% are shown in Figure 2. Table 1 shows the comparison of the proportions of MetS, its components, and HIV among patients by gender and age. A Chi-square test was used to compare the prevalence. Males showed a higher prevalence of diabetes when compared to female participants (27 vs. 32%; *p*-value < 0.001). Contrarily, compared to male participants, female individuals showed a higher prevalence of obesity and HIV positivity (*p*-value < 0.005). 

The older age group (50 years and above) showed a higher prevalence of hypertension, diabetes, obesity, and MetS when compared to the younger age group (*p*-value < 0.001). Contrarily, compared to the older age group participants, the younger age group showed a higher prevalence of being HIV positive (*p*-value < 0.001). 

The proportion of discharged alive and transferred from a primary to a referral hospital and mortality among hospitalized COVID-19 patients was 72%, 3%, 79%, and 25%, respectively (shown in Table 2). Most deaths occurred among patients aged 50 years and above constituting about 80% with almost equal numbers of those discharged alive and transferred to another facility, shown in Table 2. In terms of the district, Capricorn (20%) and Mopani (10%) had the lowest prevalence of patients being transferred to another facility as compared to other districts (about 23%). A high prevalence (above 70%) was observed among patients transferred to other facilities and those who died in the public sector observed as compared private sector ($\chi^{2}=1067.99,\;$df = 2, *p*-value < 0.001). Nonetheless, there was a minor difference in prevalence between the private (53%) and public (47%) sectors for patients who were discharged alive. Hypertensive patients comprised many COVID-19 hospitalized patients who passed away (55%) compared to non-hypertensive patients. Hypertension, diabetes, and obesity were significantly higher prevalence than non-hypertensive, diabetic, and obese, respectively (*p*-value < 0.005). Furthermore, the extended Mantel–Haenszel showed a significant linear trend for all the variables (*p* for trend < 0.005) besides for the district. Our multinomial logistic regression model’s power was satisfactory, as shown in Table 3 because it correctly identified 71.6% of the known observations and can be counted on to project future estimates. Regarding the fitted model information, the Chi-squared ratio test had a value of 317.910 (*p*-value = 0.000), indicating a good model fit. 

Only two predictors had a significant parameter for comparing the discharged alive group with the transferred to another facility group. In Table 4, the odds of being transferred to another facility were over two times higher among the ventilated than the odds of non-ventilated. A one-unit increase in the coexistence of HIV positive and MetS is associated with the decrease in the log odds of being transferred to another facility vs. discharged alive group in the amount of 10.47 holding other variables constant at zero. Furthermore, only five predictors had a significant parameter for comparing the discharged alive group with the patients who died. The odds of dying were over four times higher among patients aged 50 years and above than the odds of a patient aged less than 50 years. The odds of dying for HIV-positive patients are about 82% higher than the odds for HIV-negative. Additionally, the odds for males are about 34% higher than the odds for females. The odds of dying were over four times higher among the oxygenated than the odds of the non-oxygenated group. Holding all other factors constant at zero, a one-unit increase in ventilated patients is related to a 0.70 increase in the log odds of dying compared to the group that was discharged alive.

We further assessed the weekly mortality rate among hospitalized COVID-19 patients with different conditions, shown in Figure 3. It was observed that patients with a combination of diabetes, high blood pressure (hypertension), and obesity had a higher mortality rate after day two of admission as compared to those having a single condition (hypertension, diabetes, obesity, and HIV). In Figure 4, we assessed the independent groups visually using the survival functions graph. The different coloured lines represent the independent groups. The graph shows the cumulative survival (deaths) across the time variable (days).

It can be observed that the cumulative survival proportion appears to be much higher in the non-hypertensive and diabetic groups compared to the hypertensive and diabetic groups, respectively. In the overall comparisons using the log-rank test, the *p*-value was found to be less than 0.05 for both hypertension and diabetes-independent variables. This concludes that there is a statistically significant difference in time-to-event between the independent groups (hypertension vs. non-hypertensive and diabetic vs. non-diabetic). The hypertensive and diabetic patients diagnosed with COVID-19 significantly shortened the time until participants die compared to the other groups. There was no significant difference in time –to the event between the independent groups for obesity and MetS independent variables.

## 4. Discussion

The pathogenesis of Long-COVID is poorly understood, but this association with more severe disease, where immune dysregulation plays a major role in those with hospitalization, respiratory failure, and death, suggests an immune-mediated inflammatory dysfunction that may impacts all organs, as alluded before [53,54]. The current study identified factors and determined their relationships with hospitalization outcomes for COVID-19 patients using secondary data from the Department of Health in the Limpopo Province, South Africa. Results showed that mortality among COVID-19 patients was associated with being 50 years and older, male, and HIV positive. Furthermore, having hypertension and diabetes reduced the duration from admission to death. Being transferred to other facilities among COVID-19 patients was associated with ventilation and the coexistence of HIV plus Mets. Mets showed a higher mortality rate within seven days of hospitalization, followed by being obese. Hypertension was recorded as the lowest cause of mortality. Additionally, the spatial analysis revealed an unbalanced geographic distribution of the mortality rate, hypertension prevalence, and diabetes prevalence among hospitalized COVID-19 patients in the province of Limpopo. A high prevalence and mortality rate were recorded in the southern region of the province. However, there is scant research on the spatial distribution of diabetes and hypertension control and non-control in the province.

In this study, we found that the proportion of COVID-19 mortality (80.7%) was strongly associated with older age (≥50 years). The odds of dying were over four times higher among patients aged 50 years and above than the odds for those younger than 50 years. Not much research has been conducted in Africa to investigate the association between age and COVID-19 deaths, but several countries such as South Africa and Nigeria raised concerns over the care of older citizens who are disproportionately affected by COVID-19 [55,56]. By May 2020, the South African Medical Research Council reported that nearly 85,000 people over 60 have probably died from COVID-19 infections [56]. Consistent with our findings, a recent meta-analysis of all-cause mortality from other countries such as China, Italy, Spain, the United Kingdom, and New York State, showed that the proportion of older people who died varied across the regions [57]. In older people, severe cases of COVID-19 are characterized by acute lung injury and acute respiratory disorder syndrome (ARDS). Therefore, it is imperative to report age-specific rates of COVID-19 mortality that account for these sources of variability in the population.

We further found that males were at a higher risk to die of COVID-19 than females in this study. Their odds of dying were about 34% higher than the odds for females in Limpopo Province. For instance, in Italy, it was reported that deaths in males were approximately double compared to that of females (77.6 vs. 22.4%) [6]. Similar findings on higher mortality rates among males versus females were reported in Greece, Holland, Denmark, Belgium, Spain, China, and the Philippines [58]. The data suggest that biological differences in the immune systems between men and women impact the ability to fight infection including SARS-2-CoV-2. Additionally, it has been reported that females have a more responsible attitude toward the COVID-19 pandemic than males [59,60]. Engaging males more to improve their attitude towards COVID-19 regulations would be an effective strategy. 

Furthermore, the odds of dying among HIV-positive patients were about 82% higher than the odds for HIV-negative patients in this study. A high proportion of people newly diagnosed in South Africa are at an advanced stage of HIV infection, with a CD4 count of fewer than 200 cells/mm^3^ compromising the immune system [61]. The increased risk for severe COVID-19 disease for HIV-positive people may be attributed to immunosuppression [62], due to low CD4 cell count, advanced disease, high viral load, and those not taking antiretroviral treatment (ART) [63]. 

There are few case reports/studies on HIV/SARS-CoV-2 coinfection in Africa, except for the Ugandan study of Baluku et al. [64], which reported symptom outcomes rather than mortality in two cases. However, a study in South Africa [65] reported that 16% (22,308) of public sector patients diagnosed with COVID-19 were HIV positive, of whom 625 died. In adjusted analysis, the study found that HIV increased the risk of COVID-19 mortality. Increased HIV-associated risk of COVID-19 death remained when restricted to COVID-19 cases or hospitalized cases. The conclusion was that HIV was associated with a doubling of COVID-19 mortality risk. This was the case in countries such as Wuhan [66] and Turkey [67]. The importance of identifying co-infections and ensuring a secure supply of ART for PLHIV during the COVID-19 pandemic remains a priority. 

Pre-existing NCDs, including hypertension, diabetes, and obesity have been reported in several case-series studies among patients with confirmed COVID-19 who were hospitalized in China [68], Italy [69], and Mexico [70]. The current study showed that hypertensive patients comprised the most COVID-19 hospitalized patients who died compared to non-hypertensive patients. Hypertensive, diabetic, and obesity were significantly higher in the death group than in the transferred and discharged alive groups. 

Furthermore, having hypertension and diabetes reduced the duration from admission to death in this study. In their systematic reviews, Mirzaei and colleagues reported that co-morbidities in addition to HIV and COVID-19 (multimorbidity) included hypertension, diabetes, obesity or hyperlipidemia, and chronic obstructive pulmonary disease, while case fatality rate increased with the presence of obesity, hypertension, and/or diabetes in Mexico [70]. The presence of diabetes in COVID-19 patients was reported in other countries such as Mexico [70], the United States [71], and the United Kingdom [72]. Additionally, obesity also increases the risk of death in COVID-19 and together with diabetes results in a dysregulated immune response to respiratory infections [70]. 

We found a statistically significant difference in the time to event between patients with and without diabetes. However, the disparity in cumulative survival becomes almost equal when hospitalization is prolonged. This may be because they are treated or attended to during their admission. Diabetes-induced Microparticles (MPs) are essential in SARS-CoV-2 infection, and the virus infection that caused additional MP formation would amplify the dual function of MPs and escalate vascular inflammation in a vicious cycle, contributing to aggravated vascular dysfunction and unfavorable COVID-19 outcomes in patients with diabetes [71].

The COVID-19 pandemic has likely been substantially exacerbated by pre-existing NCDs, which might be important risk factors for severe clinical complications and outcomes in patients with COVID-19. Clustering of these NCDs and risk factors collectively called MetS [12]. MetS showed to have higher mortality within seven days of admission in this study. A systematic review and meta-analysis on MetS and its components in patients with COVID-19 [73] highlighted that patients with MetS exhibited four times greater odds of severe and fatal COVID-19 outcomes compared with those without MetS. In addition, the odds of mortality with MetS components showed an increased mortality risk as the components’ count increased, and patients with MetS had increased mortality, higher ICU admission, and increased need for mechanical ventilation [73]. This suggests that MetS should be considered a composite predictor of COVID-19 lethal outcome, increasing the odds of mortality by the combined effects of its components. 

This retrospective study has two primary limitations. Firstly, it relied on secondary data which may have affected the accuracy of the findings and the data does not contain information on vaccination status which might be important in outcome assessment. Secondly, there is a possibility of non-differential incorrect classification bias which may have resulted in an underestimation of the true relative risk. To further support and expand on these findings, it is important to conduct well-designed clinical studies. Despite these limitations, this study’s significant strength is that it is one of the few studies in South Africa and Africa that examines the coexistence of MetS and HIV in hospitalized COVID-19 patients, highlighting a higher mortality rate. These findings have important implications for clinical practice, emphasizing the need for early identification and treatment of these patients to prevent concurrent infections.

## 5. Conclusions

In this study, evidence suggests that among patients with confirmed COVID-19 who were hospitalized, older age, male gender, and MetS plus HIV might be important risk factors for severe clinical outcomes, but mostly, mortality. MetS and its components such as hypertensive and diabetes should be considered composite predictors of COVID-19 fatal outcomes. Patients with a combination of diabetes, hypertension, and obesity have a higher mortality rate after day two of admission as compared to those having a single condition (hypertension, diabetes, obesity, and HIV). Within seven days of admission, mortality rates for both MetS and obesity were higher. 

It is likely that patient outcomes will continue to be severely compromised until the problems surrounding critical care resource scarcity are addressed across South Africa. The study increases our understanding of the common contributing variables to severe manifestations and a greater mortality risk among COVID-19 hospitalized patients by investigating the influence of MetS, its components, and HIV coexistence. Nonetheless, there is limited information about the connection between (MetS) and the outcome of COVID-19 patients, and couples with HIV infection, which calls for further research. 

Prevention remains the mainstay for both communicable and non-communicable diseases, and improvement of critical care resources must be addressed across South Africa.

## Figures and Tables

**Figure 1 ijerph-20-05799-f001:**
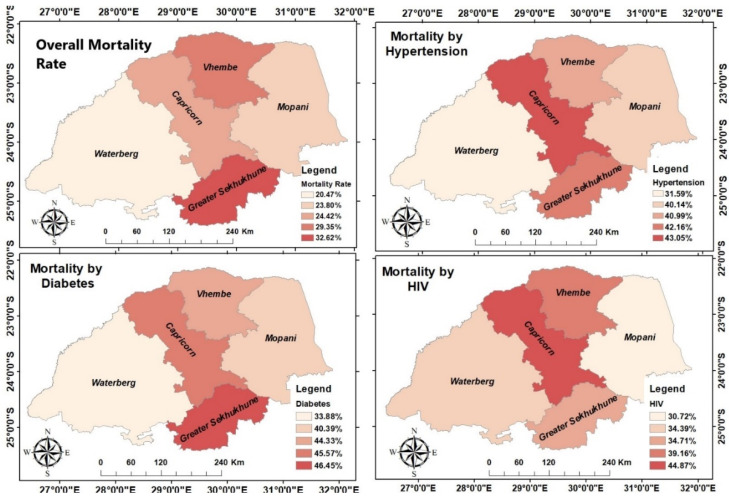
Spatial distribution of Mortality rate among hospitalized COVID-19 patients.

**Figure 2 ijerph-20-05799-f002:**
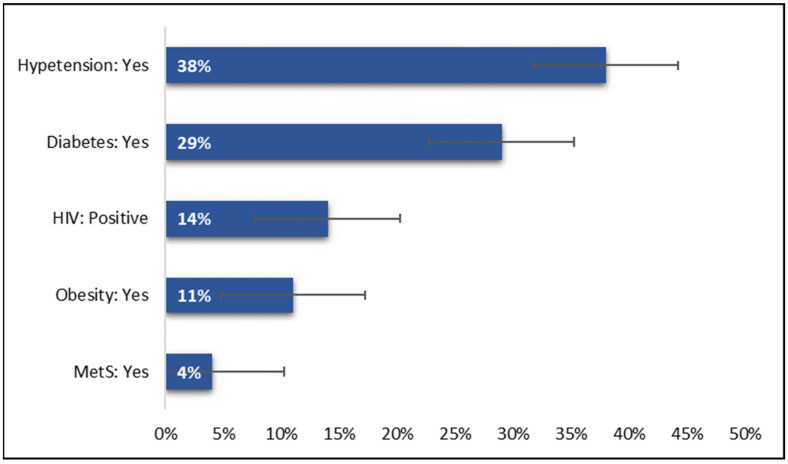
Proportions of MetS, its components and HIV.

**Figure 3 ijerph-20-05799-f003:**
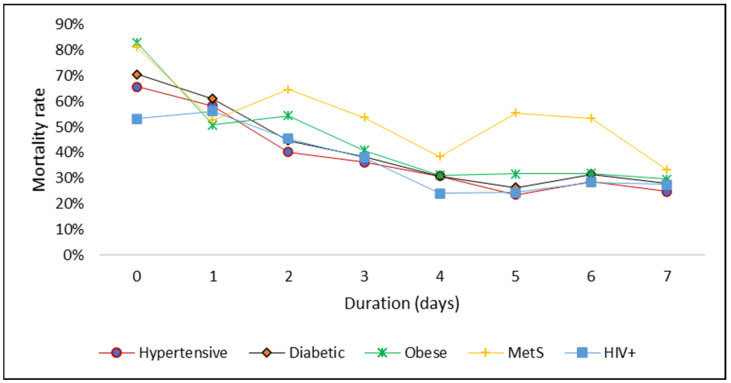
The mortality rate among hospitalized COVID-19 patients with different conditions.

**Figure 4 ijerph-20-05799-f004:**
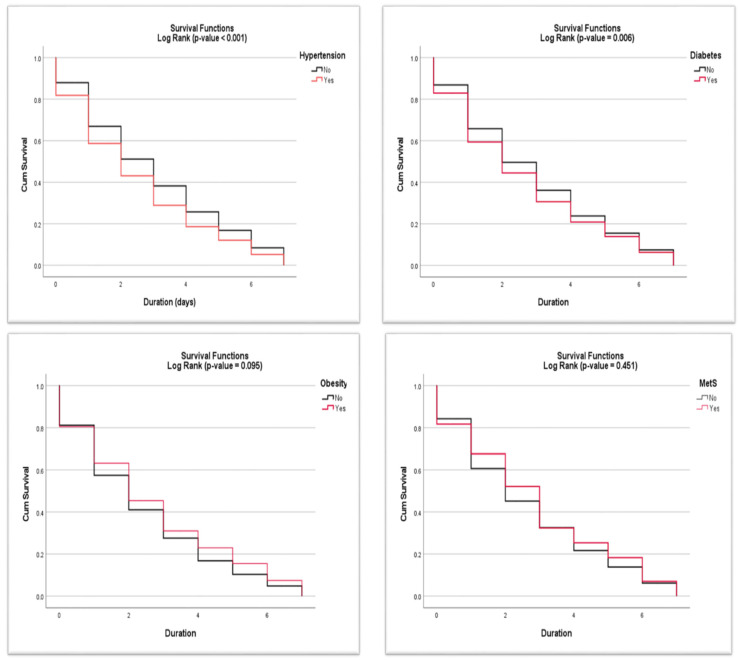
Survival curves for survival time in days among hospitalized COVID-19 patients.

**Table 1 ijerph-20-05799-t001:** Comparison of proportions of MetS, its components, and HIV among patients by gender and age.

Variables	All % (95%CI)	Males *n* = 6604 % (95%CI)	Females *n* = 8542 % (95%CI)	*p*-Values
Hypertension: Yes	38.59 (36.51; 40.71)	37.76 (34.57; 41.06)	39.17 (36.46; 41.96)	0.711
Diabetes: Yes	28.68 (26.76; 30.66)	31.70 (28.67; 34.90)	26.53 (24.11; 29.09)	<0.001 *
Obesity: Yes	11.07 (9.79; 12.50)	8.39 (6.71; 10.45)	12.98 (11.19; 14.99)	0.001 *
MetS: Yes	3.72 (2.99; 4.63)	3.38 (2.36; 4.82)	3.97 (3.00; 5.23)	0.085
HIV: Positive	13.73 (12.31; 15.29)	12.00 (9.99; 14.36)	14.96 (13.06; 17.08)	0.008 *
		<50 years *n* = 7164	≥50 years *n* = 7986	
Hypertension: Yes	38.59 (36.51; 40.71)	19.70 (17.25; 22.40)	53.70 (50.80; 56.57)	<0.001 *
Diabetes: Yes	28.68 (26.76; 30.66)	14.25 (12.14; 16.67)	40.21 (37.41; 43.08)	<0.001 *
Obesity: Yes	11.07 (9.79; 12.50)	9.25 (7.54; 11.30)	12.53 (10.74; 14.58)	<0.001 *
MetS: Yes	3.72 (2.99; 4.63)	0.87 (0.44; 1.73)	6.01 (4.77; 7.54)	<0.001 *
HIV: Positive	13.73 (12.31; 15.29)	18.06 (15.71; 20.69)	10.27 (8.64; 12.16)	<0.001 *

* *p*-value: significant at 0.05.

**Table 2 ijerph-20-05799-t002:** Frequency analysis of COVID-19-related hospitalization outcomes for patients.

Response	Discharged Alive. 10,844 (71.6%)	Transferred 468 (3.1%)	Deaths 3838 (25.3%)	*p*-Trend
Gender				<0.001 *
Female	6380 (58.8%)	250 (53.4%)	1912 (49.8%)	
Males	4460 (41.1%)	218 (46.6%)	1926 (50.2%)	
Age				<0.001 *
0–30 years	2372 (21.9%)	54 (11.5%)	110 (2.9%)	
31–49 years	3854 (35.5%)	146 (31.2%)	628 (16.4%)	
50–69 years	3496 (32.2%)	203 (43.4%)	1810 (47.2%)	
70+ years	1120 (10.4%)	65 (13.9%)	1290 (33.5%)	
District				0.052
Capricorn	4270 (39.4%)	93 (19.9%)	1410 (36.7%)	
Mopani	2094 (19.3%)	45 (9.6%)	668 (17.4%)	
Sekhukhune	985 (9.1%)	112 (23.9%)	531 (13.8%)	
Vhembe	1619 (14.9%)	109 (23.3%)	718 (18.7%)	
Waterberg	1876 (17.3%)	109 (23.3%)	511 (13.3%)	
Smoking status				<0.001 *
Current smoker	54 (4%)	6 (4%)	42 (6%)	
Former smoker	76 (6%)	5 (4%)	58 (8%)	
Never smoked	1218 (90%)	130 (92%)	607 (86%)	
Sector				<0.001 *
Private	5791 (53.4%)	124 (26.5%)	915 (23.8%)	
Public	5053 (46.6%)	344 (73.5%)	2923 (76.2%)	
Hypertensive	2035 (25.7%)	159 (49.4%)	1450 (54.8%)	<0.001 *
Diabetic	1478 (19.4%)	135 (43.7%)	1177 (47.1%)	<0.001 *
Obesity	185 (11.4%)	33 (18.2%)	174 (19.0%)	<0.001 *
MetS	47 (0.4%)	12 (2.6%)	71 (1.8%)	<0.001 *
HIV positive	509 (7.5%)	43 (24.7%)	334 (17.9%)	<0.001 *
Oxygenated	3887 (35.8%)	339 (72.4%)	2763 (72.0%)	<0.001 *
Ventilated	254 (2.3%)	42 (9.0%)	271 (7.1%)	<0.001 *

* *p*-value: significant at 0.05.

**Table 3 ijerph-20-05799-t003:** Power of the model for COVID-19 hospitalized patient outcome classification.

Observed		Predicted		Percent Correct
	Discharged Alive	Transfer	Died	
Discharged alive	206	0	30	87.28%
Transfer	14	0	1	0.00%
Died	56	0	42	42.85%
Overall percentage	79.08%	0.00%	20.92%	71.06%

**Table 4 ijerph-20-05799-t004:** The multinomial logistic regression for associations between patients’ outcomes and related sociodemographic and health factors in Limpopo Province, South Africa.

		Model 1			Model 2		
Category	Variable	B (SE)	*p*-Value	OR	B (SE)	*p*-Value	OR
	Intercept	−2.80 (0.38)	<0.0001 *	0.06	−3.66 (0.34)	<0.0001 *	0.02
Age	0–30 years	Reference		1			1
	31–49 years	0.06 (0.39)	0.8650	1.06	0.52 (0.32)	0.1090	1.69
	50–69 years	0.13 (0.41)	0.7505	1.14	1.58 (0.32)	<0.0001 *	4.85
	70+ years	−1.24 (0.80)	0.1200	0.28	2.63 (0.33)	<0.0001 *	13.95
	Male	−0.33 (0.28)	0.2502	0.71	0.29 (0.14)	0.0472 *	1.34
	Hypertensive	−0.05 (0.32)	0.8687	0.94	0.30 (0.16)	0.0610	1.36
	Diabetic	0.52 (0.33)	0.1141	1.69	0.28 (0.18)	0.1149	1.33
	HIV+	−0.90 (0.53)	0.0905	0.40	0.60 (0.21)	0.0056 *	1.82
	Obesity	−0.09 (0.46)	0.8424	0.91	−0.10 (0.27)	0.6960	0.89
	MetS	−0.34 (0.91)	0.7094	0.71	−0.01 (0.44)	0.9653	0.98
	Oxygenated	0.57 (0.31)	0.0668	1.78	1.51 (0.21)	<0.0001 *	4.53
	Ventilated	0.95 (0.46)	0.0388 *	2.59	0.70 (0.31)	0.0265 *	2.01
	HIV+ and MetS	−10.47 (<0.01)	<0.0001 *	0.01	−0.54 (0.90)	0.5615	0.58

* *p*-value: significant at 0.05; Model 1: discharged alive vs. transferred to another facility; Model 2: discharged alive and death.

## Data Availability

The dataset for the study group generated and analysed during the current study is available from the corresponding author upon reasonable request.

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
