# Peer review of "Retrospective Analysis of the Outcome of Hospitalized COVID-19 Patients with Coexisting Metabolic Syndrome and HIV Using Multinomial Logistic Regression"

_ijerph, 2023, doi:10.3390/ijerph20105799_

Round 1
Reviewer 1 Report
The manuscript entitle “Retrospective Analysis of the Outcome of Hospitalized Covid-19 Patients with Coexisting Metabolic Syndrome and HIV using Multinomial Logistic Regression“ (ID: ijerph-2312206) was reviewed carefully. The MS analyzed the coexistence of risk factors of COVID-19 appropriately. However, the MS needs some modifications such as follow:
- The English level of the MS is acceptable but it is not fluent.
- The most correct word for this viral disease is COVID-19, not Covid-19.
- For the first time in the abstract and introduction, both SARS-CoV-2 and COVID-19 must be stated in full name and then after in abbreviated form.
- Line 48: the statistics of COVID-19 worldwide must be edited in the last revised version of MS.
- Line 57: it is recommended that the paragraph to be followed with the South Africa-associated mutated variants of SARS-CoV-2 that caused the other countries to be more sensitive about travel to/from South Africa.
- In addition to association of some physiological disorders such as obesity to COVID-19, it is recommended that this association to be shown in the biochemical level such as article PMID: 32572334.
- What is the role of cytokine storm in the MetS? Check the following articles:
https://www.frontiersin.org/articles/10.3389/fimmu.2022.1015563/full
https://www.sciencedirect.com/science/article/pii/S0939475321002398#bib25
https://www.ncbi.nlm.nih.gov/pmc/articles/PMC7377208/
- Please explain the molecular reasons of affecting HIV patients with COVID-19, using:
https://www.ncbi.nlm.nih.gov/pmc/articles/PMC9258762/
https://www.ncbi.nlm.nih.gov/pmc/articles/PMC7536551/
- Line 144: blood pressure 130/85 mmHg was classified as hypertension. It must be cited. Also for diabetic and HIV patients.
- Figures contain only the title. They must contain legends as well.
- If there is a study on prevention of high mortality rate in MetS-HIV patients with vaccination, please indicate! You can check the list of possible efficient vaccines through this article:
https://www.ncbi.nlm.nih.gov/pmc/articles/PMC9968612/
- The ethical approval for this study must be checked.
Author Response
Thank you for all the comments and recommendations. The opinions were really helpful to eliminate misunderstandings and several major flows and to make the manuscript more accurate and valuable.
- The English level of the MS is acceptable but it is not fluent.
- The authors proofread the whole document to improve the language.
- The most correct word for this viral disease is COVID-19, not Covid-19.
- The authors replaced Covid-19 with COVID-19 throughout the entire document.
- For the first time in the abstract and introduction, both SARS-CoV-2 and COVID-19 must be stated in full name and then after in abbreviated form.
- Changes effected as suggested.
- Line 48: the statistics of COVID-19 worldwide must be edited in the last revised version of MS.
- Changes effected as suggested.
- Line 57: it is recommended that the paragraph to be followed with the South Africa-associated mutated variants of SARS-CoV-2 that caused the other countries to be more sensitive about travel to/from South Africa.
- Changes effected as suggested.
- In addition to association of some physiological disorders such as obesity to COVID-19, it is recommended that this association to be shown in the biochemical level such as article PMID: 32572334.
- Changes effected as suggested.
- What is the role of cytokine storm in the MetS? Check the following articles:
https://www.frontiersin.org/articles/10.3389/fimmu.2022.1015563/full
https://www.sciencedirect.com/science/article/pii/S0939475321002398#bib25
https://www.ncbi.nlm.nih.gov/pmc/articles/PMC7377208/
- Thank you for the comment, references were added as suggested.
- Please explain the molecular reasons of affecting HIV patients with COVID-19, using:
https://www.ncbi.nlm.nih.gov/pmc/articles/PMC9258762/
https://www.ncbi.nlm.nih.gov/pmc/articles/PMC7536551/
- Thank you for the comment, references were added as suggested.
- Line 144: blood pressure 130/85 mmHg was classified as hypertension. It must be cited. Also for diabetic and HIV patients.
- Thank you for the comment, authors added citations to those categories.
- If there is a study on prevention of high mortality rate in MetS-HIV patients with vaccination, please indicate! You can check the list of possible efficient vaccines through this article:
https://www.ncbi.nlm.nih.gov/pmc/articles/PMC9968612/
- Thank you for asking this question; it would be interesting if we had this data in our dataset. However, the data didn't exist because this was essentially a surveillance system that merely tracked the number of infected and those who passed away without accounting for treatment. So we were not able to do a comparison among patients vaccinated.
- The ethical approval for this study must be checked.
- Thank you for the comment, the authors added the Ethical number by the Turfloop Research Ethics Committee (TREC).
Reviewer 2 Report
Thank you for the opportunity to review this article.
Comment:
-Please include in the discussion comments about the difference of the results between the districts (related with poor controled diabetes and hypertension or not), eventually based on te data of the population MetS and HIV positive (8 million patients)
- Please reformulate line 108 ("covid disease"), lines 179, 180 (the presentation of every disease percentage and correct the Table 4 to Table 1)
Author Response
Thank you for all the comments and recommendations. The opinions were really helpful to eliminate misunderstandings and several major flows and to make the manuscript more accurate and valuable.
-Please include in the discussion comments about the difference of the results between the districts (related with poor controled diabetes and hypertension or not), eventually based on te data of the population MetS and HIV positive (8 million patients)
- Thank you for the comment, the authors added the discussion comments about the difference in the results between the districts.
- Please reformulate line 108 ("covid disease"), lines 179, 180 (the presentation of every disease percentage and correct the Table 4 to Table 1)
- Thank you for the comment, the authors effected the changes as suggested.
Reviewer 3 Report
Good study with quite large number of patients.
Title can be improved to make is concise is possible.
I have the following concerns and questions.
1. A lot of inconsistent use of "COVID-19" os covid19 or Covid19. Please use standard format.
2. spelling errors with SARS-CoV2 (line 92)
3. A thorough literature search is always encouraged for the intro. Citing papers supporting your hypothesis strengthens your paper. The following paper was missing in your review of lit:
https://journals.physiology.org/doi/full/10.1152/ajpheart.00409.2022
4. In figure 1, please mention that the mortality %ages depicted are in COVID-19 patients. It appears like the map depicts mortality due to each of the conditions only- like hypertension, HIV etc.
5. Please improve figure legends. They are too vague and not descriptive enough.
6. In table 3, all the numbers add up nicely. Any reason that the total discharged alive is 10844 instead of 10840?
7. Figure 4: what is the x axis- duration ? only mentioned days in one graph.
8. The survival difference for diabetes is visible in the early days, but becomes almost same later on. Any explanation for this? You may want to explain this in the text as well because in the text there is a simple conclusion that there is significant difference in survival.
9. Line 239-247. This paragraph is very confusing. It seems the authors are not clear which figure (3 or 4) they are referring to in each line. The statements contradict the figures. please revise this paragraph and be specific which figure is being referred to clearly.
10. Figure 4 refers to survival curve in weeks or days? There seems to be inconsistent use of the duration times.
Author Response
Thank you for all the comments and recommendations. The opinions were really helpful to eliminate misunderstandings and several major flows and to make the manuscript more accurate and valuable.
I have the following concerns and questions.
- A lot of inconsistent use of "COVID-19" os covid19 or Covid19. Please use standard format.
- Thank you for the comment, the authors changed Covid-19 to COVID-19 throughout the entire document.
- spelling errors with SARS-CoV2 (line 92)
- Thank you for pointing that up; the error has been fixed.
- A thorough literature search is always encouraged for the intro. Citing papers supporting your hypothesis strengthens your paper. The following paper was missing in your review of lit:
https://journals.physiology.org/doi/full/10.1152/ajpheart.00409.2022
- Thank you for the comment, the authors added this missing paper to strengthen the paper.
- In figure 1, please mention that the mortality %ages depicted are in COVID-19 patients. It appears like the map depicts mortality due to each of the conditions only- like hypertension, HIV etc.
- Thank you for the comment, changes were effected as suggested.
- Please improve figure legends. They are too vague and not descriptive enough.
- Thank you for pointing that up, however, from the ArcGIS, this is the most accurate figure legend we could find.
- In table 3, all the numbers add up nicely. Any reason that the total discharged alive is 10844 instead of 10840?
- Thank you for the comments, in some categories, such as gender, there was some missing data, whereby 4 cases was lacking information.
- Figure 4: what is the x axis- duration ? only mentioned days in one graph.
- Thank you for the question, x-axis on the survival curve always represent the cumulative survival probability which is the probability of survival.
- The survival difference for diabetes is visible in the early days, but becomes almost same later on. Any explanation for this? You may want to explain this in the text as well because in the text there is a simple conclusion that there is significant difference in survival.
- Thank you for the question, the authors tried to explain the survival difference for diabetes in the early days and later on.
- Line 239-247. This paragraph is very confusing. It seems the authors are not clear which figure (3 or 4) they are referring to in each line. The statements contradict the figures. please revise this paragraph and be specific which figure is being referred to clearly.
- Thank you for pointing that up, the authors deleted the sentence that made the paragraph unclear.
- Figure 4 refers to survival curve in weeks or days? There seems to be inconsistent use of the duration times.
- Thank you for pointing that up, Figure 4 refers to survival curve in days as highlighted in the caption.
Reviewer 4 Report
Dear Authors,
congratulations on your valuable work. Please, find below some suggestions that, in my humble opinion, could be useful to further improve the quality of your paper.
1. Please, pay attention to the spelling of the acronyms (you mentioned "Covid-19" but the correct wording is "COVID-19").
2. Please, be careful with some statements. The citation referred to the sentence "In Italy, the 60 most prevalent comorbidities among patients with confirmed Covid-19 were hypertension and diabetes" is quite outdated. Please, please check if this is still true or if there are any updates.
3. I think it is important, 3 years after the start of the pandemic, to mention how certain risk factors can also change the therapeutic approach to COVID-19 infection. In this sense, I recommend updating the introduction with some references also to the efficacy of antiviral drugs and monoclonal antibodies in these patients, so as to provide the reader with a more complete overview. Below you can find some references to this regard (but there are many more in the literature):
- doi: 10.1007/s40121-022-00729-2
- doi: 10.3390/v15020384
- doi: 10.1007/s10238-023-01036-x
- doi: 10.1002/jmv.28660
- doi: 10.1016/j.clinthera.2022.01.007
4. It is not made explicit in the analysis whether your patients had received vaccination for COVID19 and/or whether they had received specific treatment. I think it is important in outcome assessment to be aware of this information and to stratify the analysis according to this as well.
Author Response
Thank you for all the comments and recommendations. The opinions were really helpful to eliminate misunderstandings and several major flows and to make the manuscript more accurate and valuable.
- Please, pay attention to the spelling of the acronyms (you mentioned "Covid-19" but the correct wording is "COVID-19").
- Thank you for the comment, the authors changed Covid-19 to COVID-19 throughout the entire document.
- Please, be careful with some statements. The citation referred to the sentence "In Italy, the 60 most prevalent comorbidities among patients with confirmed Covid-19 were hypertension and diabetes" is quite outdated. Please, please check if this is still true or if there are any updates.
- Thank you for the comment, changes were effected as suggested. There was still true.
- I think it is important, 3 years after the start of the pandemic, to mention how certain risk factors can also change the therapeutic approach to COVID-19 infection. In this sense, I recommend updating the introduction with some references also to the efficacy of antiviral drugs and monoclonal antibodies in these patients, so as to provide the reader with a more complete overview. Below you can find some references to this regard (but there are many more in the literature):
- doi: 10.1007/s40121-022-00729-2
- doi: 10.3390/v15020384
- doi: 10.1007/s10238-023-01036-x
- doi: 10.1002/jmv.28660
- doi: 10.1016/j.clinthera.2022.01.007
- Thank you very much for the comment. The study, according to the authors, was more focused on the epidemiological element than it was on the treatment because the data available was scarce.
- It is not made explicit in the analysis whether your patients had received vaccination for COVID19 and/or whether they had received specific treatment. I think it is important in outcome assessment to be aware of this information and to stratify the analysis according to this as well.
- Thank you for asking this question; it would be interesting if we had this data in our dataset. However, the data didn't exist because this was essentially a surveillance system that merely tracked the number of infected and those who passed away without accounting for treatment.
Round 2
Reviewer 3 Report
The authors have addressed all the concerns. Therefore, I recommend the paper be accepted.
Author Response
Our sincere thanks to our reviewer for their excellent comments and suggestions which improved the presentation.
Reviewer 4 Report
Dear Authors,
thank you for accepting part of my suggestions. I am still convinced that it could be further improved by adding some consideration.
I see that you do not have any information about the vaccination status of you patients and I understand why but I think that you might include this as a study limitation, at least.
I do not agree with your comment on the third point of my first round revision. Risk factors drive therapies and I think you should add something about this topic in the introduction.
Thank you very much for your work and consideration,
Best regards
Author Response
We thank the reviewer for restating these points, which we hope we to have now addressed by adding the information about vaccination status under the limitation section and citing suggested references in line 117-124.
The opinions were really helpful to make the manuscript more accurate and valuable.